# Correlates of Social Isolation in Forensic Psychiatric Patients with Schizophrenia Spectrum Disorders: An Explorative Analysis Using Machine Learning

**DOI:** 10.3390/ijerph20054392

**Published:** 2023-03-01

**Authors:** Lena Machetanz, Steffen Lau, David Huber, Johannes Kirchebner

**Affiliations:** Department of Forensic Psychiatry, Psychiatric Hospital, University of Zürich, 8032 Zurich, Switzerland

**Keywords:** forensic psychiatric patients, offending, schizophrenia spectrum disorder, social isolation, psychosocial burden

## Abstract

The detrimental effects of social isolation on physical and mental health are well known. Social isolation is also known to be associated with criminal behavior, thus burdening not only the affected individual but society in general. Forensic psychiatric patients with schizophrenia spectrum disorders (SSD) are at a particularly high risk for lacking social integration and support due to their involvement with the criminal justice system and their severe mental illness. The present study aims to exploratively evaluate factors associated with social isolation in a unique sample of forensic psychiatric patients with SSD using supervised machine learning (ML) in a sample of 370 inpatients. Out of >500 possible predictor variables, 5 emerged as most influential in the ML model: attention disorder, alogia, crime motivated by ego disturbances, total PANSS score, and a history of negative symptoms. With a balanced accuracy of 69% and an AUC of 0.74, the model showed a substantial performance in differentiating between patients with and without social isolation. The findings show that social isolation in forensic psychiatric patients with SSD is mainly influenced by factors related to illness and psychopathology instead of factors related to the committed offences, e.g., the severity of the crime.

## 1. Introduction

Social integration is well known to be a fundamental need in humans as social species with a beneficial impact on physical and mental health [1,2]. Social contacts may promote certain health-related behaviors, work through direct psychosocial mechanisms (i.e., social support and control), and even positively influence physiological processes in well-integrated individuals, such as biochemical stress responses and cardiovascular functions [3,4,5]. In turn, social isolation, defined as small social networks, infrequent social contacts, the absence of confidante connections, living alone, and lack of participation in social activities, has shown to be a psychosocial burden of serious magnitude [5,6].

People suffering from mental illness are particularly susceptible to social isolation [7]. Specifically high rates of social isolation have been reported for forensic psychiatric patients, who can be considered to be dual-vulnerable regarding social isolation due to their mental illness and their involvement with the criminal justice system [8]. This stems partially from stigmatization but also from an impairment of complex psychosocial and cognitive functions required for social interaction, such as mentalizing or bond/relationship forming [9,10].

With a prevalence of 50–60%, individuals with a diagnosis of schizophrenia spectrum disorder account for the majority of patients within forensic psychiatric measures [11,12]. SSD are known to severely affect social functioning as they cause deficits in numerous domains relevant to social functioning, such as emotion recognition, emotion processing, and theory of mind [9]. It comes as no surprise that there is a high rate of social isolation in populations with SSD, with a recent study reporting rates of 80% [13].

The social burden of a lack of societal integration does not stay without consequences: for patients suffering from a schizophrenia spectrum disorder (SSD), a positive association between social isolation and depressive and negative symptoms as well as impaired social and cognitive functioning has been shown [14,15,16]. Loneliness has also been found to be associated with increased use of maladaptive coping strategies, higher levels of self-blaming, and lower self-esteem in patients with SSD [13]. In fact, there is also subtle evidence of a reciprocal influence of social isolation and schizophreniform symptomatology: findings from murine animal models strongly support the hypothesis that social isolation leads, in fact, to deficits associated with SSD, such as anxiety and cognitive impairments [17,18]. Apart from its negative effect on mental and physical health, social isolation also correlates with criminal and violent behavior [19]. Therefore, tackling social isolation has been the focus of recovery-orientated treatment approaches not only in general psychiatry but also in forensic psychiatric care [20,21,22].

While the population of forensic psychiatric patients with SSD is considered to be at high risk for social isolation due to their twofold disadvantage described above, there is little research on protective and risk factors. To close this research gap, the present study aims to exploratively evaluate factors associated with social isolation in a unique sample of forensic psychiatric patients with SSD using machine learning.

## 2. Materials and Methods

Our population consisted of 370 male and female forensic psychiatric patients with a schizophrenia spectrum disorder (F2x acc. to ICD-10) who had been in court-mandated inpatient treatment between 1982 and 2016 at the Centre for Inpatient Forensic Therapies of the University Hospital of Psychiatry Zurich, Switzerland. The majority of case files stemmed from treatments from the year 2000 on (296 cases). Offences leading to the referenced forensic psychiatric hospitalization included both violent crimes, including (attempted) homicide, assault, violent offences against sexual integrity, robbery, and arson, and non-violent crimes, including threats and coercion, property crime without violence, criminal damage, traffic offences, drug offences, and illegal gun possessions. This patient population has been evaluated regarding other aspects, e.g., inpatient aggression, using a similar methodologic approach, which is why extracts of the following section may be replicated in part [23,24,25,26].

Data collection was performed retrospectively from the patients’ case files, which comprised professionally documented patient history, psychiatric/psychologic inpatient and outpatient reports, extensive reports from clinicians as well as nursing and care staff, testimonies, court proceedings and data regarding the offence(s) leading to the referenced forensic hospitalization and, where applicable, previous imprisonments and detentions.

Data extraction in the form of a directed qualitative content analysis was carried out by a psychiatrist with experience in forensic psychiatry using a standardized rating protocol, which had been based on a set of criteria proposed by Seifert et al. and extended in cooperation with other senior researchers in forensic and general psychiatry [27,28,29]: the dataset derived from the case files included items from the following categories: socio-demographic data, childhood/youth events, psychiatric history, criminal history, psychosexual functioning, prison data, particularities of the current hospitalization as well as psychopathological symptoms defined by an adapted three-tier positive and negative syndrome scale (PANSS) [30]. For a detailed description and definition of all predictor variables, please refer to our coding protocol in the data availability statement.

A second independent clinician encoded a random subsample of 10% of all cases to evaluate for inter-rater reliability, which was considered substantial with a Cohen’s kappa of 0.78 [31].

As the aim of this study was to exploratively identify the variables with the most influence out of numerous possibly relevant parameters, supervised ML was applied. In contrast to unsupervised ML, which is used to discover hidden patterns in unlabeled datasets, supervised ML trains algorithms on labeled datasets and uses these algorithms to predict defined outcomes [32].

Figure 1 and Figure 2 provide an overview of the statistical procedure step-by-step. All steps were performed using R (v 3.6.3) and the MLR package v2.171. CI calculations of the balanced accuracy were conducted using MATLAB R2019a (MATLAB and Statistics Toolbox Release2012b, The MathWorks, Inc., Natick, Massachusetts, United States) with the add-on “computing the posterior balanced accuracy” v1.0 [33,34].

The first step was to preprocess the data for ML (see Figure 1, Step 1). Variables with >33% of missing values were eliminated from further analysis. All categorical variables were converted to binary code, while continuous and ordinal variables underwent no adjustment. The outcome variable “*social isolation at the time of the offence*” was defined according to Tanskanen et al. and was considered “present” if the patient had suffered from small social networks, infrequent social contacts, the absence of confidante connections, living alone, and lack of participation in social activities for a period of at least 1 year before admission to the referenced forensic hospital [5]. The variable was considered “true” if at least 3 of these aspects were met. After data preparation, the dataset was split into two subsets: one training subset consisting of 70% of all cases and one validation subset consisting of the remaining 30% (Figure 1, Step 2). The validation set was stored aside for the application of the trained algorithm, while the training subset was used during the learning process. Further steps (Figure 1, Step 3a–c) were applied to the training subset only. Missing values in the predictor variables were imputed by mean for continuous variables and by mode for categorical variables using the features applied in the MLR package (Figure 1, Step 3a). As provided by the “impute”-function included in the MLR package, an “ImputationDesc” object was created, which contained the coefficients used in the imputation on the training set. This allowed the application of the same coefficients in the imputation of missing values in the validation set, following at a later stage in this process.

The outcome variable was upsampled due to the uneven distribution of “social isolation” (70.6% v. 29.4%, see results), thus leading to a more balanced outcome (Figure 1, Step 3b). To spare computational resources and increase the overall performance of the model, we reduced the number of variables through the application of a random forest algorithm (randomForestSRC package implemented in the MLR package, evaluating variable importance) (Figure 1, Step 3c). This reduction in dimensionality was performed up to the point where the AUC did not improve by >5% by adding another variable. Following these preprocessing procedures, seven different algorithms were applied to the training set for discriminative model building: logistic regression, trees, random forest, gradient boosting, k-nearest neighbor (KNN), support-vector machines (SVM), and naïve Bayes as an easily applicable generative model. These algorithms were assessed in terms of their balanced accuracy (the average of the true-positive and true-negative rate) and goodness of fit (measured with the receiver operating characteristic, balanced curve area under the curve method, ROC-balanced AUC). We also evaluated specificity, sensitivity, positive predictive value (PPV), and negative predictive value (NPV). Then, the model with the highest AUC was selected for final model validation (Figure 1, Step 4): Variables underwent testing for multicollinearity to avoid dependencies between the variables. To avoid overfitting, a common problem in ML that occurs when the algorithm corresponds exactly to the training data, including the incorporation of outliers, a cross-validation process strictly separate from the validation of the model is advisable. For this purpose, we performed a nested resampling: data processing and model training were performed embedded in cross-validation, and the performance of these models was tested in an outer loop also embedded in cross-validation. Thus, we artificially created different subsamples of the same dataset, all while the validation subset remained untouched (Figure 1, Step 5). This was followed by model building and testing on the validation subset previously split from the training subset (see Figure 2).

In the first step, the imputation of missing values was performed using the same weights as on the training set (Figure 2, Step 1). Then, the most suitable model, which was identified in Figure 1, Step 4, was applied and evaluated in terms of its performance parameters (Figure 2, Step 2). Lastly, all identified predictor variables were ranked according to their indicative power within the model (Figure 2, Step 3).

## 3. Results

After the exclusion of 50 cases due to missing data on the independent variable, a total population of 320 patients remained. Out of these, 226 (70.6%) were socially isolated at the time of the offence leading to the referenced forensic psychiatric hospitalization, while the remaining 94 (29.4%) were not. As expected within a judicial system, the population was predominantly male (90.9%) and early middle-aged (mean: 34 years). Three-fourths of the cases were also unmarried at the time of the offence. Table 1 provides an overview of the basic population characteristics and their distribution among the two groups.

### 3.1. Model Building through ML

After applying 7 different algorithms, naïve Bayes emerged as the one with the best performance parameters on the training set (see Table 2).

Table 3 shows an overview of the 5 variables that emerged as most dominant in the model out of the over 500 possibly influential variables.

### 3.2. Applying the Model

When applied to the validation subset, with which the algorithm had not come in contact yet, naïve Bayes yielded a balanced accuracy of 69.2% and an AUC of 0.74. Both patients who were socially isolated and patients who were not socially isolated were identified correctly in the majority of cases (Table 4).

### 3.3. Predictor Variables Regarding Social Isolation: Influence in the Model

After ranking the predictor variables according to their relative influence in the model, the items referring to symptoms at the time of the offence leading to the referenced forensic psychiatric hospitalization—*attention disorder* and *alogia*—emerged as most influential, followed by *crime motivated by ego disturbances*, the *total score of the adapted PANSS upon admission,* and a *history of negative symptoms* (see Figure 3).

## 4. Discussion

Our explorative study identified the five most dominant predictors of social isolation in a homogenous sample of forensic psychiatric patients with a schizophrenia spectrum disorder, a population that makes up a large portion of patients hospitalized in forensic psychiatric institutions and is thus highly relevant for forensic psychiatric research. As expected in a population with a severe mental disorder and a criminal background, the majority of patients were subjected to social isolation upon their admission to the referenced hospitalization. In the model-building process, naïve Bayes emerged as the most suitable algorithm for the dataset. The final model yielded a balanced accuracy of 69.2% and an AUC of 0.74, which can be considered substantial. With a specificity of 79% and a sensitivity of 60.7%, the model was able to correctly identify patients subjected to social isolation in 4/5 of the cases. Interestingly, amongst over 500 variables possibly influential to the model, the most relevant items that emerged were all related to psychopathology.

Most prominent was *attention disorder at the time of the offence* leading to the referenced hospitalization. Attention in the sense of comprehension and/or concentration is commonly found to be heavily impaired in patients with SSD [35]. In the literature, there is a well-documented correlation between the ability for social perception and socio-functional outcome measures, especially problem-solving, social behavior in the milieu, and functioning within a community [36]. Naturally, the ability to correctly process information as part of social cognition is necessary to generate appropriate reactions in interactive situations [37]. In turn, the inability to perceive direct information as well as indirect information through social cues, such as intonation or posture of the counterpart, critically impairs patients’ capacity to interact with others [38]. It seems, therefore, unsurprising that this item emerged as most predictive regarding social isolation not only in patients with SSD in general but also in this subpopulation of forensic psychiatric patients with SSD. The second most relevant item, *alogia at the time of the offence*, hits a similar notch. It seems self-explanatory that alogia, a symptom domain that can be split into poverty of speech, latency of response, blocking, and poverty of the speech content, leads to an impairment of social interaction simply due to the reduced capacity for expression [39]. A recent study, which also applied machine learning in order to model alogia, reported correlations between reduced social functioning and the presence of alogia [40]. It has also been hypothesized that attention disorders and alogia—as the two most influential items in this study—are, in fact, interlinked. Researchers have found associations between speech production and attention as well as other cognitive functions and have even suggested that these cognitive deficits may actually be the cause of alogia [41].

It seems noteworthy to consider that one is looking at a chicken-and-egg situation with the identified items. Reddy et al., for example, showed a decrease in (social) cognitive functioning in study participants with SSD after social exclusion [15]. It could be hypothesized that social isolation may not only be consequential to cognitive impairments in patients with SSD but may, in fact, be causal or at least contributing. Here, further research regarding causality is yet pending.

*Crime motivated by ego disturbances* emerged as the third most influential factor in the model. Ego disturbances, defined according to Kurt Schneider as “permeability of the ego-world boundary”, are known to be linked to deficits in social cognition, so their influence in a model dominated by social cognition impairment is unsurprising [42,43]. It seems quite logical that the ability to discriminate between self and other is necessary in order to correctly understand the subjective experiences of others and that, in turn, the inability to do so impairs social interaction. This conclusion is supported by neuroscientific publications describing a sense of self and self-recognition as having developed in humans to model the internal worlds of others, thus allowing them to infer intentions and causes that lay behind observed behaviors and therefore improving the efficiency of social interaction [44]. However, the question remains why not simply ego disturbances but ego disturbances as a motivational factor for the committed offence emerged as a dominant factor. The authors hypothesize that perhaps individuals whose ego disturbances become actionable suffer from a more severe expression of this symptom, thus being more hindered in their ability to interact socially.

The *Total PANSS score upon admission* was identified as the fourth most influential factor. Patients subjected to social isolation showed a higher adapted PANSS score than their socially integrated counterparts; the more pronounced the severity of the mental illness, the more isolated the affected individual. Social isolation has been described to affect all kinds of symptom domains: Negative emotions such as stress and anxiety caused by the burden of social isolation may trigger or intensify hallucinations and paranoia, an observation that was also made in healthy individuals [45,46]. As described above, social isolation also leads to a decrease in cognitive function [47]. Michalska da Rocha et al. described the relationship between social isolation and SSD symptoms as a “self-perpetuating cycle of exclusion” [46]. They hypothesized that the schizophrenia spectrum disorder impairs the ability to socially interact and maintain relationships, leading to a breakdown of important protective factors (e.g., corrective and supportive structures), thus increasing the likelihood of escalation of psychotic episodes, further social withdrawal, and so forth. Such a reciprocal relationship would well explain why the severity of the symptomatological expression was so dominant in the model. However, while such findings are rather robust across different studies, experimental rather than observational studies are essential in order to facilitate a better understanding of the linkage between social isolation and an increase in symptomology as well as the direction of this relationship.

Finally, the model was influenced by a *history of negative symptoms*. Negative symptoms, defined as loss of certain, otherwise normal functions, have been identified before as predictors of poor quality of life, negatively affecting social functioning and interpersonal relationships [48]. Several pathways between negative symptoms and social isolation can be inferred. Anhedonia, the reduced capability to derive pleasure from interaction, may lead to less affiliative feelings of closeness with others, while avolition and asociality can lead to a decreased willingness to interact with other people and consequently reduce the ability to function in communities [49,50]. As described above, alogia, as a component of negative symptoms, significantly impairs the ability to communicate with others [39]. An impoverishment of facial expressions also hinders social performance skills [51].

To sum up our findings, forensic psychiatric patients with SSD subjected to social isolation are characterized in particular by domains affecting social cognition, including attention disorder, alogia, and actionable ego disturbances, overall psychopathology as measured by the PANSS, and a history of negative symptoms.

Interestingly, items related to criminal history, e.g., the severity of the offence leading to the referenced forensic hospitalization or the number of previous convictions and incarcerations, did not emerge as highly influential in the model for discriminating between patients with and without social integration. This matches findings from one of the authors’ previous publications exploring predictors of social integration after a court-mandated forensic psychiatric treatment: here, too, the variables most predictive of social isolation after discharge from inpatient treatment related to antisocial behavior and the patients’ living conditions before the forensic psychiatric hospitalization, while the patients’ criminal background, such as the number of previous incarcerations, played only a small role in the model [52]. This goes to show that the development and maintenance of social ties are not so much hindered by juridical problems but that other domains, especially psychopathology, are more dominant.

Another notable aspect was that, similar to items from the domain of criminal history, biographical and psychosocial items also had no major indicative power within the model. This is surprising, as, for example, disruptive life events, unemployment or migration experiences are generally known to be predisposing to social isolation, at least within the general population [53,54]. However, it is quite likely that for people suffering from a severe mental illness, items related to psychopathology become overbearing compared to environmental or biographical items.

Limitations. The present analysis was conducted exclusively based on retrospectively extracted data, which particularly complicates the collection of parameters that are difficult to define, such as hostility. The retrospective design also possibly affected the outcome variable, as social isolation was not assessed using a psychometric scale or other instrument but only deducted from information from the clinical records. Therefore, it was not possible to assess the degree and patients’ subjective perception of social isolation. Previous authors have described distinct differences between the effects of objectifiable social isolation and a feeling of loneliness [5,55]. Therefore, it would have been valuable to evaluate whether patients with no social network did, in fact, subjectively suffer from this and to evaluate both items separately.

While the large quantity of explored variables (>500 items) made machine learning a suitable approach, the rather small population was not ideal for statistical purposes. The performance of machine learning models in recognizing patterns within a dataset is proportional to its size—the smaller the dataset, the less accurate the algorithms are [56]. While we tried to counteract this issue through preprocessing and the application of appropriate algorithms, the model should be validated in larger samples in order to draw robust conclusions regarding causal inferences. As mentioned briefly at the beginning of our discussion, this research cannot answer the question of causality—whether social isolation predates the dominant items or whether it is their consequence remains unanswered.

Lastly, as to be expected in a population with a history of criminal behavior, our sample consisted mainly of male subjects, thus limiting the applicability to female forensic psychiatric patient populations. However, we decided not to exclude the few female patients in order to depict the real gender relations in the mass and penal system.

In summation, our findings demonstrate that illness-related factors are the most dominant domain among a large quantity of variables in discriminating between forensic psychiatric patients with SSD suffering from social isolation and those who do not. This sparks hope, as, in contrast to static items such as previous incarcerations, all of those items can be influenced therapeutically: through sufficient treatment of the underlying SSD, patients can be empowered to form and maintain social ties, while untreated symptoms aggravate social isolation. Nevertheless, a cautionary aspect remains when the results are put into context with the findings of a previous study by the authors mentioned above [52]: when the population was evaluated for predictors of social ties after discharge from court-mandated treatment, only 37.9% of all patients had some sort of social network upon their release. Upon their admission, the very same patient population had social ties in 29.4%, meaning that only roughly 10% of all patients could actually form and/or maintain a social network during their hospitalization even though their hindering symptoms were treated. While this is understandable, as being institutionalized in a highly monitored and secured setting obviously leaves little room for building and maintaining social contacts, it suggests that symptom remission alone does not lead to social re-integration and highlights the importance of the incorporation of other interventions aiming at personal recovery.

## 5. Conclusions

Social isolation is known to have detrimental effects on mental and physical well-being. It is also associated with criminal behavior, thus presenting not only a burden for the affected individual but also for society. The present findings facilitate a better understanding of factors associated with social isolation in a particularly vulnerable group: forensic psychiatric patients with SSD as a severe mental illness. The authors consider this population to be highly relevant in forensic psychiatric research as a majority of patients hospitalized in forensic psychiatric institutions are affected by some kind of psychotic disorder. Through the application of sophisticated statistical methods (supervised machine learning), we identified the 5 items most related to social isolation out of over 500 possible predictor variables. All of these predictors stemmed from the domain of psychopathology. The findings of this explorative study point to the importance of symptom management: tackling cognitive deficits, negative symptoms, and overall psychopathology seems crucial in the reintegrative recovery process, thus lifting the burden of social isolation in forensic psychiatric patients with SSD. Since there is growing evidence for a reciprocal relationship between symptomatology and social isolation, future research is needed to contribute to a comprehensive understanding of the underlying mechanisms.

## Figures and Tables

**Figure 1 ijerph-20-04392-f001:**
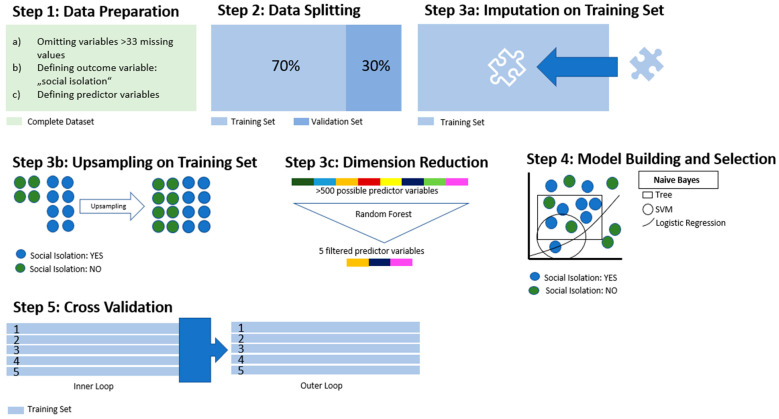
Statistical procedures: data processing and training.

**Figure 2 ijerph-20-04392-f002:**
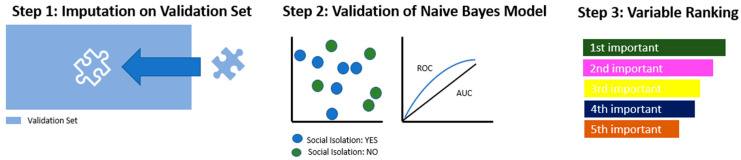
Statistical procedures: model building and testing on validation set.

**Figure 3 ijerph-20-04392-f003:**
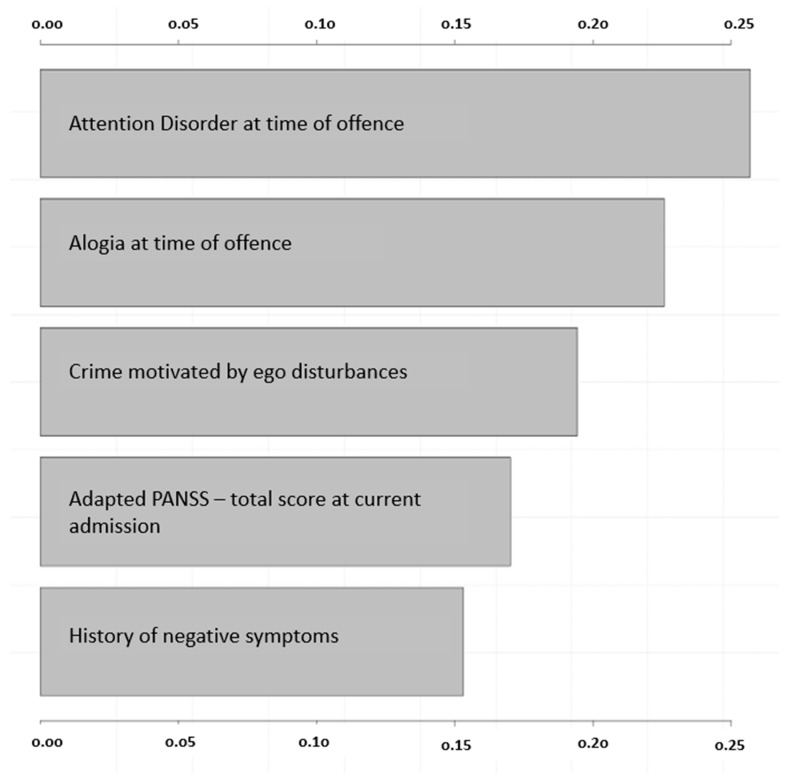
Importance of variables in the naïve Bayes model.

**Table 1 ijerph-20-04392-t001:** Basic characteristics of the study population.

Characteristics	Total n/N (%)	Social Isolation n/N (%)	No Social Isolation n/N (%)
Sex: male	291/320 (90.9)	205/226 (90.7)	86/94 (91.5)
Age at admission (mean, SD)	34.3 (10.4)	34.7 (10.0)	33.2 (11.2)
Native country: Switzerland	158/320 (49.4)	114/226 (50.4)	44/94 (46.8)
Single (at offence)	260/318 (81.3)	186/225 (82.7)	74/93 (79.6)
Diagnosis: schizophrenia	264/320 (82.5)	189/226 (83.6)	75/94 (79.8)

SD = standard deviation; N = total study population; n = subgroup with characteristic.

**Table 2 ijerph-20-04392-t002:** Applied machine learning algorithms and their performance in nested cross-validation.

Algorithm	Balanced Accuracy (%)	AUC	Sensitivity (%)	Specificity (%)	PPV (%)	NPV (%)
Logistic Regression	65.7	0.77	63.60	67.80	45.90	82.50
Tree	67.8	0.70	63.50	72.10	47.90	82.50
Random Forest	65.6	0.76	64.9	66.2	43.7	81.9
Gradient Boosting	67.7	0.76	65.7	69.7	48.9	83.5
KNN	64.4	0.73	60.9	67.9	44.5	81.1
SVM	67.8	0.75	62.4	73.1	49.5	82.8
Naive Bayes	70.1	0.80	75.1	65.1	46.4	85.1

AUC = area under the curve (level of discrimination); sensitivity = true positive/(true positive + false negative); specificity = true negative/true negative + false positive; PPV = positive predictive value; NPV = negative predictive value. KNN = k-nearest neighbors; SVM = support-vector machines.

**Table 3 ijerph-20-04392-t003:** Absolute and relative distribution of most dominant predictor variables.

Variable Description	Social Isolation n/N (%)	No Social Isolation n/N (%)
History of negative symptoms	166/225 (73.8)	43/94 (45.7)
Alogia at time of offence	66/154 (42.9)	4/45 (8.9)
Attention disorder at time of offence	89/152 (58.6)	11/44 (25)
Adapted PANSS—total score at current admission	25.1 (SD = 13.5)	20.1 (SD = 10.9)
Crime motivated by ego disturbances	127/225 (56.4)	27/94 (28.7)

N = total study population; n = subgroup with characteristic; SD = standard deviation; PANSS = positive and negative syndrome scale.

**Table 4 ijerph-20-04392-t004:** Performance parameters of naïve Bayes model on the validation set.

Performance Parameters	% (95%-CI)
Balanced accuracy	69.2 (64.4–81.7)
AUC	0.74 (0.64–0.84)
Sensitivity	60.7 (60.1–61.3)
Specificity	79.4 (79.1–79.7)
PPV	54.8 (54.3–55.4)
NPV	83.1 (82.8–83.4)

CI = confidence interval; AUC = area under the curve (level of discrimination); sensitivity = true positive/(true positive + false negative); specificity = true negative/true negative + false positive; PPV = positive predictive value; NPV = negative predictive value.

## Data Availability

The dataset generated and analyzed during the current study is available from the corresponding author upon reasonable request. A detailed list of all our variables (including definitions and references) is available under the following link: https://www.researchgate.net/publication/363044110_Coding_protocol_Pathways_into_delinquency_in_offenders_suffering_from_schizophrenia_spectrum_disorders (access date: 30 August 2022).

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
