# Peer review of "Correlates of Social Isolation in Forensic Psychiatric Patients with Schizophrenia Spectrum Disorders: An Explorative Analysis Using Machine Learning"

_ijerph, 2023, doi:10.3390/ijerph20054392_

Round 1
Reviewer 1 Report
Correlates of social isolation in offender patients with schizophrenia, spectrum disorders and explorative analysis using machine learning
ijerph-2166939
Overall this paper provides an interesting way of examining the correlates associated with social, isolation and individuals, who are mandated by the court to inpatient care.
The main difficulty I have with this paper is the extensive use of jargon without corresponding definitions to the vast majority of methods used to compile and analyze the data. For me, it makes what was done by the researchers nearly illegible and the need to take it on faith that what they did was appropriate. For example, the construction of the data was done through contact analysis. Yet the authors never defined what texts they were using to generate data and then collate data on each patient. Additionally, what sorts of themes the coders applied were not discussed. Additionally, there was a discussion of imputation. I understand that the authors imputed means and modes for the missing values in variables, however, what was the technique used to do that? Additionally, weights were applied at some stage in the model, and were never described how they were created, and why they were used. When you get to the machine learning element of the paper we are never told what type of prediction is being done. My suspicion is that it is a logit model with an ROC curve given the statistics coming from the analysis, but it would be better if the others told us what they did. It’s implied in figure 2, but not explicitly described. In table 3 they mention absolute and relative distribution, without a description of what those numbers mean, nor do they explain specificity and sensitivity in table 4. In short, this project folds in everything from qualitative content analysis and the decision-making involved to traditional coding multiple imputation, weighting data and machine learning and predictor testing. I recognize the authors don’t have a lot of space to discuss these things, but I can’t follow many of the decisions the authors made because of lack of explanation surrounding them. Also, machine learning and data science approaches in social sciences are just taking hold. It’s unreasonable to assume that your reader will understand these terms.
Additionally, I take issue with the term offender patient. Offender is a stigmatizing term one that we don’t commonly use in criminology anymore when talking about individuals, particularly in correctional spaces which are involved in this paper. The term forensics psychiatric patients which is used on the first page might be more suitable.
Finally, there are a few places in the paper that need some copy editing. For example, a sentence might be missing an “a” or an abbreviation was used.
Lastly, I really liked the right up of predictive variables in the discussion section. The authors is really contextualize their findings within the literature. The only aspect that I think could be buttressed in the discussion as why this matters for individuals with schizophrenia spectrum disorders most of the discussion focuses on treatment of the characteristics relating to social isolation, but outside of a few sentences in the introduction I don’t see a deep discussion of my social isolation is critical to stamp out for individuals with SSD.
Author Response
Dear Sir or Madam, please find our responses to your queries in the document attached.

Reviewer 2 Report
Thank you for the opportunity to review this paper on correlates of social isolation in people with offense histories who are diagnosed with schizophrenia spectrum disorders. Your method and results are clearly described and the discussion provides a nice interpertation of your findings. I think that the introduction section is the weakest part of the manuscript.
In the introduction fairlly limited attention is paid to schizophrenic spectrum disorders. Indeed, there is a link between psychotic symptoms and social isolation but why do you specifically focus on this population? Many other disorders are associated with social isolation. In fact, most forensic patients deal with this, sometimes because of their offense (e.g., patients with sexual offense histories are often stigmatized and find it difficult to maintain a social network). Was it a matter of convenience to look at SSD or did you have a rationale for this? (e.g., high prevalence of SSD in forensic populations, higher risk of recidivism etc. ).
In the abstract, you state that 6 topics emerged as most influential but you only mention 5. In the result and discussion section, also 5 topics are presented/discussed so may be it's a typo in the abstract?
Author Response
Dear Sir or Madam,
please find our responses to your queries in the attachment.

Reviewer 3 Report
In this study the authors investigated the use of machine learning techniques to evaluate the correlates of social isolation in forensic patients affected by schizophrenia spectrum disorders. Overall, this is an interesting study, the manuscript is well written, the statistical analysis is appropriate, and the references are up-to date.
However, I have some concerns:
In the Introduction, line 40 please substitute “consequences: For example” with “consequences: for example”
In the Materials and Methods section the authors state: “criteria proposed by Seifert et al. and extended in cooperation with other senior researchers in forensic and general psychiatry”. Can you be more precise and describe them briefly?
In my opinion, it is a bit unclear how the authors measured the outcome “social isolation”, have this been done simply by reading the clinical records? In this case, I think that this may constitute a limit and should be discussed, as it would have been better to use a psychometric scale.
In addition they state: “The outcome variable “social isolation at the time of the offence” was defined acc. to Tanskanen et al. and was considered “present” if…”Can you be more precise? Must all the component be present to define the patient as “Socially isolated” or 4 o 3 out of 5 can be enough?
In the Discussion section, line 166, please put a “.” after “psychopathology.
In the Discussion section, line 247, please substitute “psychiatric treatment: Here” with “psychiatric treatment: here”. In addition, please rephrase this sentence “Here, the most predictive variables related to life conditions before the patient’s forensic psychiatric hospitalizations and antisocial behavior rather than the patients’ criminal background”, it is not clear to me.
In the Limitation section, line 262, please substitute “variable: It was” with “variable: it was”
In the Limitation section, line 285, please substitute “therapeutically: Through” with “therapeutically: through”
In the Conclusions section, line 307, please substitute “deduced: Tackling” with “deduced: Tackling”
Author Response
Dear Sir or Madam,
please find our responses in the attachment.

Round 2
Reviewer 1 Report
Thank you for your changes to the document. I feel like readers would have a far better understanding of your methods and procedures and be able to replicate your methods in their own work. Well done with such a constraint in space.